# Forecasting Research on Urban Green Development Based on System Dynamics—A Case Study of Hefei in China

Yanling Feng [1,*] , Boqiang Liu [1], Qiang Yan [1] and Guozhu Jia [2]

1. School of Modern Post (School of Automation), Beijing University of Posts and Telecommunications, Beijing 100876, China; 2023111363@bupt.cn (B.L.); yan@bupt.edu.cn (Q.Y.)
2. School of Economics and Management, Beihang University, Beijing 100191, China; jiaguozhu@buaa.edu.cn
* Correspondence: yanling@bupt.edu.cn

**Abstract:** Urban green development is a way of economic growth and social development aiming at efficiency, harmony and sustainability, and in recent years urban green development has become an important trend for future urban development. In this study, Hefei City is selected as the study area, and a comprehensive green-development model is constructed by applying the system dynamics method, which integrates five important subsystems, namely, economy, environment, people's livelihood, S and T and resources. Through simulation analysis, this study reveals the dynamic trend of green development in Hefei City. The analysis results show that economic growth provides the foundation for green development, scientific and technological innovation promotes the development of green technology, and that sustained investments in people's livelihoods and environmental protection improves people's quality of life and a city's eco-friendliness. This study aims to promote the coordinated economic, social and environmental development of Hefei to ensure that Hefei is steadily moving towards high-quality and sustainable development goals; it effectively reveals the current situation, forecasts the trend of green development in Hefei, and also provides a reference for the urban green development of other cities.

**Keywords:** urban green development; economic growth; system-dynamics model; simulation analysis

## 1. Introduction

In China's "13th Five-Year Plan", green development was proposed. "Made in China 2025" and the "Industrial Green Development Plan" also pointed out that solving environmental and resource constraints in China's economic development could be done through the implementation of green development. Green development, as a key strategy for realizing comprehensive sustainable development, has become a common goal pursued globally and plays an important role in finding a balance between promoting economic growth and protecting the environment in particular. According to previous studies, China's development history can be traced back to a stage of development that has gone through the stages of disorder, circularity and sustainability, and is in the process of transitioning to the concept of green development [1,2]. This transformation not only responds to the constraints of the natural environment, but also solves the ecological problems encountered in economic development and meets the needs of the people for an excellent ecological environment, as well as addressing the poor quality of the ecological environment [3,4]. In addition, the concept and practice of green development is also a core component of China's current "supply-side reform", which is of great significance in promoting the transformation of a healthy growth model for the economy, especially under the current new economic normal [5]. With the gradual disappearance of the labor dividend, the development model of single reliance on factor inputs has become insufficient, thus accelerating the transformation of the development model, and focusing on the improvement of quality and efficiency has become the key to future development [6,7].

The concept of urban green development has evolved since it was first introduced in 1987 in the report Our Common Future, and has continued to evolve in subsequent academic and policy discussions. By 2005, the United Nations Economic and Social Commission for Asia and the Pacific (UNESCAP) defined green growth as an environmentally sustainable approach to economic growth, while in 2007 the United Nations Environment Programme (UNEP) defined a green economy as one that creates decent jobs and values nature [8]. In the wake of the 2008 financial crisis, there has been a global exploration of a new paradigm of green economy and a green new deal to address the challenges of sustained growth and sustainable development [9]. In 2011, OECD proposed that green development is not only a way to achieve economic growth, but is also a key to preventing environmental degradation and the overuse of resources [10]. The Rio+20 Conference in 2012 further strengthened the global consensus on green development and promoted the diversification of green economic models [11].

Chinese researchers (Huang, J.H. et al.) [12–14] have conducted extensive studies exploring the implementation of green-development strategies, covering concepts, paths, strategies and the drivers and constraints of green development, and have developed a more systematic theoretical framework. Feng Tao et al. [3] discovered that recyclability, low carbon and sustainability are the core concepts of urban green development and the key to improving the efficiency of green development. Subsequent researchers and scholars have found that this efficiency improvement is reflected in the decoupling of economic growth from resource consumption and pollution emissions through improved resource utilization and the promotion of green science and technology innovations [15–19]. Relevant studies have used the DEA model, the directional distance function model and the total factor productivity model to assess the efficiency of green development, and at the same time explored its spatial and temporal evolution characteristics through Malmquist exponential decomposition and spatial autocorrelation methods [20–22]. Lin Peng and Meng Nana [23] measured the green-development efficiency of six major urban agglomerations using a three-stage super-efficiency SBM-Malmquist-Luenberger productivity-index model, and examined the spatial and temporal divergence of the growth of green-development efficiency using the Dagum Gini coefficient method and the dynamic spatial SDM model. In addition, relevant studies have focused on the impact of factors such as fiscal decentralization, technological innovation and environmental regulation on the efficiency of green development [24,25]. These empirical research results provide theoretical and methodological support for practicing green development in China, as well as a detailed analysis of green development in different regions and industries.

However, although existing studies provide a rich theoretical and methodological foundation in the field of green-development efficiency, there is still a lack of comprehensive research on specific cities such as Hefei under the framework of the coupled system of economy–society–nature, and no scholars have used the system-dynamics approach to forecast the urban green development of Hefei City. The innovation of this study is that this study fills this gap by constructing and simulating a green-development system-dynamics model for Hefei, providing a systematic approach to assess and analyze the efficiency of urban green development. By integrating multiple subsystems such as economy, science and technology, environment, livelihood and resources, this study not only reveals the current status and forecasts the trend of Hefei's urban green development in various fields, but also makes specific recommendations for Hefei's future green-development strategies against domestic and international research progress.

The rest of this study is organized as follows: Section 2 introduces the basic theoretical structure of our model and the system-dynamics approach of key subsystems in the model. Section 3 demonstrates the validation of the model. Before modeling-decision analysis, it is necessary to validate the model to verify its scientific validity, and then proceed with modeling-decision analysis. Section 4 analyzes the simulation results of the system-dynamics model of green development in Hefei City. Section 5 discusses the main results of

the simulation through scenario analysis. Section 6 provides conclusions and suggestions for future research.

## 2. System Dynamics Methodology

Forecasting is a decision-making system in which past decisions depend on predictions about the future [26]. The demand of forecasting processes is complex due to various economic, social and technological factors and their inter-relationships. System dynamics (SD) helps to model, simulate and analyze the nonlinear behavior of complex phenomena over time, and is well suited for forecasting complex systems under uncertainty [27–29]. Sustainability models provide more reliable short- and long-term trend projections than statistical models and allow for decision makers to identify plausible scenarios to inform decision making and policies [30]. Jo, C. et al. used system-dynamics methods to predict the future unemployment and employment rates in South Korea [31]. Li, Y. et al. used system-dynamics models as modeling techniques, constructing an innovation- and policy-simulation model for Tianjin's green-development system [32]. The autocorrelation approach performs economic analysis by implementing feedback mechanisms in a nonlinear dynamical model consisting of stocks and flows. The advantage of systems dynamics is to think about the problem in a logical rather than numerical way. Compared to classical prediction methods such as machine learning, systems thinking provides an interpretability of simulations and avoids the formation of black-box models, and is now widely used in problem analysis, decision-making advice, academic communication, model training and other fields.

Many researchers used the system-dynamics approach to solve urban-development problems. Arasteh, M.A. and Farjami, Y. [33] developed a system-dynamics-based and agent-based simulation-based water-resource economic model to enhance the sustainability of urban groundwater. Tan, Y. et al. [34] evaluated the sustainability level of the city through a system-dynamics model and conducted simulation analysis on the case study of Beijing, helping policy makers formulate relevant strategies and achieve better urban sustainable development. Wang, H. and Bao, C [35] evaluated the historical situation of the Beijing Tianjin Hebei urban agglomeration by constructing an ecological security index and a system-dynamics model, and predicted future scenarios, providing a framework for evaluating the historical and future status of ecological security. Ani, M.S. et al. [36] used a comprehensive system-dynamics model to develop a comprehensive toolset to evaluate the potential impact of measures taken by decision makers on sustainable urban transportation and analyzed the transformation path of sustainable urban transportation. Pluchinotta, I. et al. [37] used a system-dynamics model to examine how multiple interventions (e.g., socio-environmental and economic policy incentives) can reduce drinking-water consumption by urban residents in Ebbsfeldt Gardens City. Hu, W. et al. [38] evaluated the potential impact of an urban underground passenger-transportation network (URFT) on the externalities and internal operational performance of urban freight transportation using system-dynamics methods, and provided scientific references for the selection of similar green-logistics measures and promoted logistics innovation in large cities. Zhao, P. et al. [39] developed an integrated two-stage dynamical model of the power system to quantitatively assess the environmental impacts of the integrated development of urban subsurface resources, including the integrated development of subsurface space and subsurface materials. O'Keeffe, J. et al. [40] utilized a system-dynamics model to quantify and evaluate the impact of London's urban design on the relationship between the natural and built environments. Relevant researchers have adopted a system-dynamics approach to simulate and compare the evolutionary states of different resilience subsystems within the city, and to explore the resilience mechanisms of the city at different stages of development [41,42]. Norouzian-Maleki, P. et al. [43] combined a system-dynamics approach with data-envelopment analysis techniques, focusing on the need for managers to identify the most effective policies to manage travel demand in urban congestion.

In this paper, we adopt the system-dynamics approach to build a model to study the current situation of urban green development in Hefei and to forecast the trend of future development, aiming to promote the coordinated development of economy, society and environment in Hefei and to ensure the steady progress towards the goal of high-quality and sustainable development, and at the same time to adjust the relevant parameters of the system-dynamics model, which puts forward a specific proposal for the future green-development strategy of Hefei, and also provides a reference for the urban green development of other cities.

### 2.1. Model Boundary and Basic Assumptions

In delving into the main drivers of green development in Hefei City, this study includes the economy, science and livelihood, the environment, livelihood and resources within the content boundary of the system. By integrating existing studies, this study constructs a framework for urban green development consisting of five subsystems: the economic subsystem, the science and technology subsystem, the environmental subsystem, the livelihood subsystem and the resource subsystem.

The year 2010 is selected as the base year of the model, the simulation step is set to one based on experience, and the time boundary of the system is set to 2010–2030. Among them, 2010–2022 is the modeling time period, which is mainly used for the historical test to verify the validity of the constructed model; 2022–2030 is the prediction time period of the model, which is mainly used for the prediction of the future state of the green development in Hefei City.

On the basis of clarifying the purpose of the study and the system boundary, in order to ensure the scientific and intuitive nature of the model, this study puts forward the following seven assumptions:

1. The urban green development of Hefei City is a continuous and gradual process;
2. In the model analysis, we will ignore unconventional factors such as force majeure and significant changes in the external environment;
3. Scientific and technological innovations are able to simultaneously improve production efficiency and reduce energy consumption and emissions;
4. Improvements in people's livelihoods will increase social recognition and participation in green development, creating a virtuous cycle;
5. Government policies play a key guiding and supervisory role in green development;
6. Infrastructure and urban planning are adapted to the requirements of green development;
7. Increased public awareness of environmental protection will promote green consumption behaviors, which will in turn drive innovation in green products and services by enterprises.

With the above assumptions, this study aims to create a comprehensive system-dynamics model to scientifically analyze and forecast the future trend of urban green development in Hefei City.

### 2.2. Causality Diagrams

This study demonstrates the dynamic process of urban green development through a causality diagram, as shown in Figure 1; the main feedback loops are marked with black lines: ① GDP → +industrial output → +solid-waste generation/wastewater generation/ exhaust-gas generation → -environment quality → +GDP. ② GDP → +GDP per capita → +exhaust-gas generation → -environment quality → +GDP. ③ GDP → +financial expenditure → +transportation expenditure → +vehicle ownership for road operations → +exhaust-gas generation → -environment quality → +GDP. ④ GDP → +financial expenditure → +Expenditure on education → +public-library holdings → +number of patents in the population → +GDP. ⑤ GDP → +financial expenditure → +expenditure on education → +number of students enrolled in general higher education → +R&D staff → +number of patents in the population → +GDP. ⑥ GDP → +financial expenditure → +S and T expenditures → +R&D staff → +number of patents in the population → +GDP.

⑦ Total water consumption → -water supply-and-demand ratio → +total water resources → +modulus of water yield → +demographic → +residential water consumption → +total water consumption. The feedback loops ④, ⑤ and ⑥ are positive feedback loops, and they have a reinforcing effect. And the feedback loops ①, ②, ③ and ⑦ are negative feedback loops, as they have a balancing effect. The whole system covers five core subsystems: economy, science and technology, environment, livelihood and resources. Each subsystem and its interactions are described in detail below.

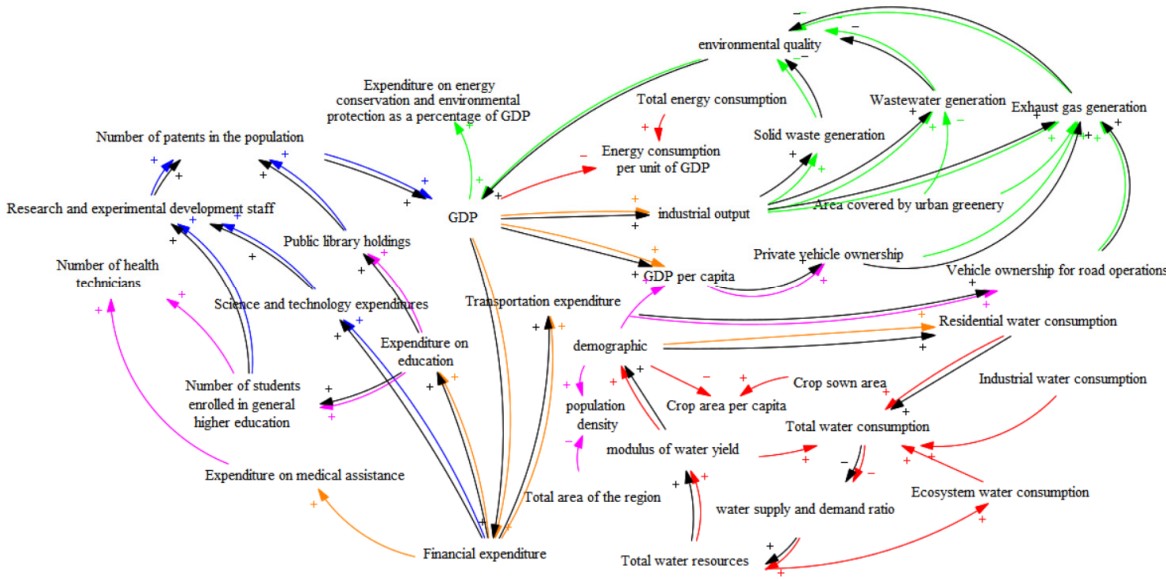

**Figure 1.** Results of causality diagram of the urban green-development system.

Economic subsystem: The economic subsystem is an indicator that assesses the level of development of a city through GDP. A city's GDP is positively correlated with government spending, meaning that an increase in GDP will lead to an increase in spending in areas such as healthcare, education, science and technology and transportation, which is marked by an orange path in the causal diagram.

S and T subsystem: At the heart of the S and T subsystem is S and T spending, which is directly linked to the number of students in higher education institutions and the size of the research workforce. An increase in the number of researchers predicts an increase in the number of patents granted per 10,000 people, thus contributing to an increase in scientific and technological progress and innovation capacity, which is represented by the blue path.

Livelihoods subsystem: The livelihoods subsystem emphasizes growth in GDP per capita and elevated transportation expenditures, which will lead to an increase in the number of vehicles operating on roads. Similarly, elevated Medicaid spending will increase the number of health technicians serving in hospitals. Increased spending on education encourages more students to pursue higher education and join the healthcare system after graduation, as well as increased investment in public libraries, which increases the number of books in the collection, and these relationships are marked by purple paths.

Environmental subsystem: The environmental subsystem is reflected in the emissions of wastewater, exhaust gases and solid wastes resulting from the growth in industrial output, which have a direct impact on the quality of the environment. At the same time, the increase in vehicles operating on the highway will also exacerbate the emissions. The role of urban greening, on the other hand, plays a positive role in absorbing exhaust gases and purifying wastewater, which is shown as a green path.

Resource subsystem: The resource subsystem focuses on the impact of population growth and economic development on urban resources. In the resource subsystem, the relationship between water resources, food resources and energy resources and the economy and population are analyzed one by one. The water-resources demand of Hefei City

is mainly composed of residential water, industrial water and ecological-environment water, so the total water consumption in the system flow diagram is composed of three parts. Energy consumption is inseparable from economic growth, so it is related to GDP growth. Food resources are related to the sown area of crops, so the link between crop area and population is established. Energy-saving and environmental-protection expenditures are added to the resource subsystem to further measure the relationship between the environment and resources in Hefei. It is shown as a red path.

### 2.3. Stock Flow Diagram

According to the causality diagram, the flow stock diagram is drawn to construct the system-dynamics model of urban green development, as shown in Figure 2. After constructing the system-dynamics model of urban green development, the key step in the empirical analysis stage is data collection and processing. For this reason, this study takes Hefei City as an example, and systematically collects the time series data of relevant indicators between 2010 and 2021. The data source is mainly from the Hefei City Statistical Yearbook, which provides an official and authoritative database for the study. Through the data-collection and processing steps, this study ensures the coherence and reliability of the data used, which prepares us for the next step of constructing the equation formulas between the variables, and also provides a solid data foundation for the simulation and analysis of the model (https://www.hefei.gov.cn/mlhf/sj/nd/, accessed on 1 October 2023).

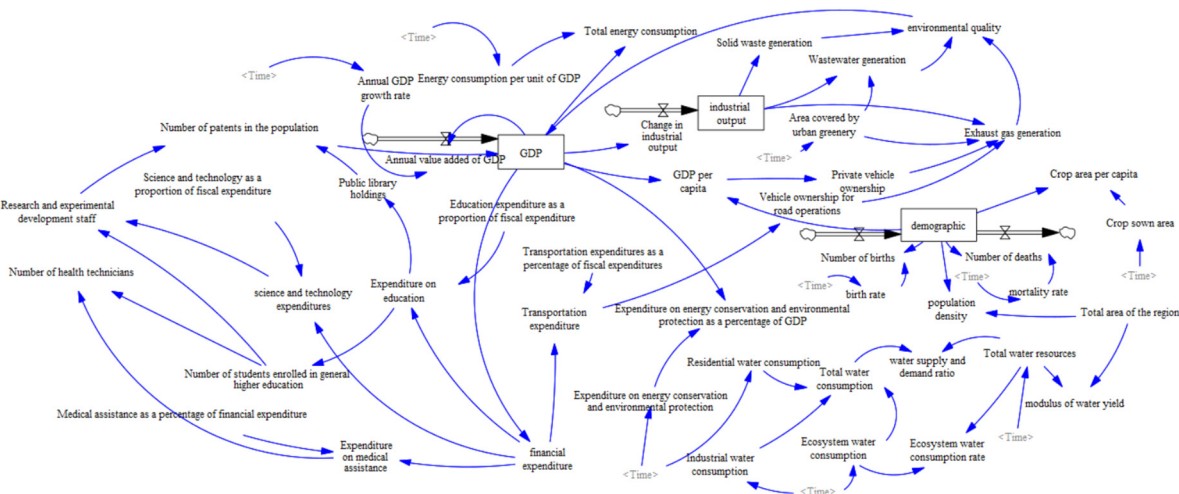

**Figure 2.** Results of stock flow diagram of Hefei's urban green-development system.

### 2.4. Constructing Variable Equations

Based on the collected data, various methods such as a literature method, a table-function method and a regression-equation-fitting method were used to explore the functional relationship between the variables in the urban green-development system. In this case, the literature method refers to reviewing the results of previous research to find out whether previous research has been conducted on this subject; the table-function method refers to the description of variable relationships through the table function in Vensim software 10.0.1; the regression-equation-fitting method refers to the selection of multiple equations to fit the relationship between the variables, and the curve fitting selected for this study includes SPSS linear regression, polynomial fitting, exponential fitting, Gaussian fitting and logistic regression. From these curve fittings, the highest fit was selected as the regression equation to describe the variable relationships. Partial fits are shown in Figure 3a for the annual change in industrial value added,Figure 3b for wastewater generation and Figure 3c for solid-waste generation.

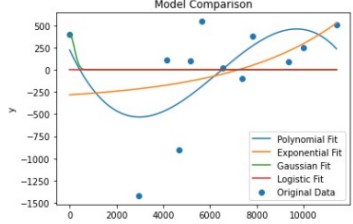

(**a**) Fitting of annual change in industrial value added

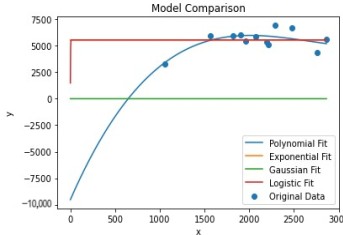

(**b**) Fitting of wastewater generation

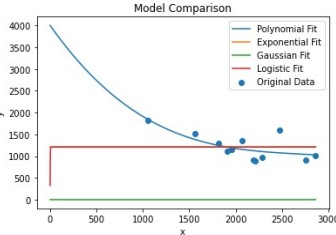

(**c**) Fitting of solid-waste generation

**Figure 3.** Partial fits in the regression-equation-fitting method.

As a result, the functional relationships of the variables in the urban green-development system are summarized, and Table 1 shows the relevant formulas of the key variables involved in the urban green-development system model of Hefei City. The variables involved in all the subsystems of the specific system and the related descriptions are shown in Table A1 (see Appendix A).

**Table 1.** Correlation formulae for key variables.

| Variant | Formula |
|---|---|
| GDP | GDP = INTEG (Annual value added of GDP, 2961.67) |
| Annual value added of GDP | Annual value added of GDP = GDP $\times$ Annual GDP growth |
| Annual GDP growth | Annual GDP growth = WITH LOOKUP (Time, ([(0, 0)–(2022, 0.2)], (2010, 0.175), (2011, 0.154), (2012, 0.1361), (2013, 0.1221), (2014, 0.1), (2015, 0.105), (2016, 0.098), (2017, 0.085), (2018, 0.0854), (2019, 0.0787), (2020, 0.0434), (2021, 0.092), (2022, 0.035))) |
| Industrial output | Industrial output = INTEG (Change in industrial output, 1052.71) |
| Change in industrial output | Change in industrial output = $100 \times \text{GDP}^{1.8}$ |
| Financial expenditure | Financial expenditure = $170 \times \text{GDP}^{1.253}$ |
| Number of health technicians | $4.779 \times 10^{-6} \times$ Expenditure on medical assistance + $2.974 \times 10^{-6} \times$ Number of students enrolled in general higher education + 0.905 |
| Demographic | Demographic = INTEG (Number of births − Number of deaths, 745.7) |
| Birth rate | Birth rate = WITH LOOKUP (Time, ([(0, 0)–(20, 10)], (2010, 0.1109), (2011, 0.108), (2012, 0.126), (2013, 0.1188), (2014, 0.131), (2015, 0.134), (2016, 0.164), (2017, 0.1994), (2018, 0.1695), (2019, 0.1356), (2020, 0.1287), (2021, 0.0978), (2022, 0.0897))) |
| Mortality rate | Mortality rate = WITH LOOKUP (Time, ([(0, 0)–(20, 10)], (20100, 0.0434), (2011, 0.0597), (2012, 0.0797), (2013, 0.0754), (2014, 0.0611), (2015, 0.053), (2016, 0.0468), (2017, 0.095), (2018, 0.0468), (2019, 0.0384), (2020, 0.0509), (2021, 0.048), (2022, 0.0539))) |
| Number of patents granted to the population | Number of patents granted to the population = $-0.0209624 \times$ Research and experimental development staff − $0.0125769 \times$ Public-library holdings + 115202 |
| Research and experimental development staff | $-0.0209624 \times$ Number of students enrolled in general higher education − $0.0125769 \times$ Expenditures on science and technology + 115202 |
| Solid-waste generation | Solid-waste generation = $-1.30771 \times 2.718 \times$ industrial output$^3$ + $1.06979 \times 2.718 - 3 \times$ industrial output$^2$ + $0.972946 \times 2.718 + 3$ |
| Exhaust-gas generation | Exhaust-gas generation = $0.45 \times$ Vehicle ownership for road operations + $30.565 \times$ Area covered by urban greenery − $19.999 \times$ industrial output − $0.193 \times$ Private vehicle ownership − 291695 |
| Wastewater generation | Wastewater generation = $2.718 \times 9.64643 - 7 \times$ industrial output$^3$ − $7.71311 \times 2.718 - 0.3 \times$ industrial output$^2$ + $1.93174 \times 2.718 +$ industrial output $-9.52156 \times 2.718 + 3 -$ Area covered by urban greenery |

**Table 1.** *Cont.*

| Variant | Formula |
|---|---|
| Area covered by urban greenery | Area covered by urban greenery = WITH LOOKUP (Time, ([(0, 0)–(2021, 30,000)], (2010, 12,737), (2011, 14,804), (2012, 15,334), (2013, 16,683), (2014, 18,428), (2015, 19,072), (2016, 19,477), (2017, 20,115), (2018, 22,893), (2019, 23,382), (2020, 23,851), (2021, 25,195))) |
| Total water consumption | Total water consumption = Industrial water consumption + Residential water consumption+ Ecosystem water consumption |
| Total energy consumption | Total energy consumption = GDP × 10000 × Energy consumption per unit of GDP |
| Crop-area per capita | Crop area per capita = Crop sown area/(demographic × 10000) |
| Total water resources | Total water resources = WITH LOOKUP (Time, ([(0,0)–(2021, 10)], (2010, 30.12), (2011, 28.33), (2012, 30.99), (2013, 29.44), (2014, 49.69), (2015, 49.76), (2016, 87.83), (2017, 37.64), (2018, 54.31), (2019, 21.52), (2020, 89.15), (2021, 51.06)) |

## 3. Model Check

### 3.1. Operational Check

The runtime test actually includes two parts: the quantitative check and the model structure check, and the main purpose of the model structure test is to check whether the model is complete and whether it can run normally. The most frequently reported error in this test is that the internal equation is not assigned a value or the relationship with other variables is not constructed. Using the model check function of "Vensim 10.0.1" software, the results show that there is no problem with the model. Since the model uses standardized data with a uniform scale, the scale check is reasonable.

### 3.2. Stability Check

The stability check is conducted using industrial output as an example to observe the stability of the system under different simulation time intervals. Change the time step, Test 1 "TIME STEP = 0.5" and Test 2 "TIME STEP = 1.5" to compare and analyze with the initial current "TIME STEP = 1" to determine whether the system simulation model is stable or not. As shown in the Figure 4, although there are slight differences in the industrial output at different time intervals, they all show a steady upward trend, and the trend showing that the system behavior is consistent, so the model is stable.

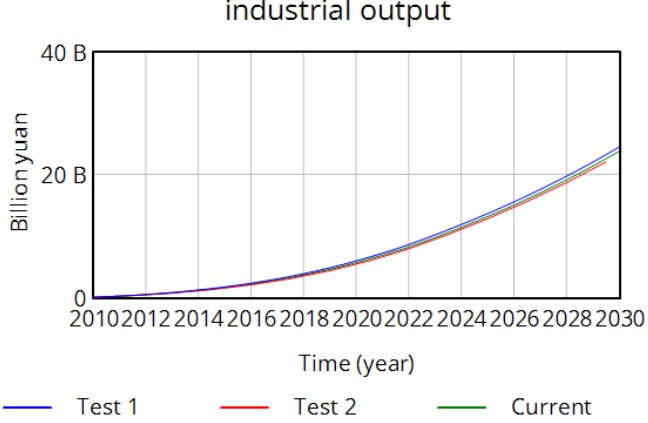

**Figure 4.** Stability comparison results with different time steps.

### 3.3. Historical Check

The historical check is an effective means to verify whether the model is reliable, only when the error between the simulation value and the real value is small; it can show that the model is accurate and objective, and can predict well the future data. However, it is

difficult to objectively reflect the future status of the research object. In this paper, GDP, demographic and total water resources are key variables selected for the historical check, the test time is 2010–2021, and the test results are shown in Table 2. As can be seen from Table 2, the error between the historical data and the simulation value of the three index is less than 10%, indicating that the constructed system model and the real system have a better fit and smaller deviation, which can reflect the development status of the city to a large extent, and can effectively analyze and forecast the future development level of the urban green city of Hefei City.

**Table 2.** Results of the historical check of the model.

| Index | GDP (CNY Billion) | | | Demographic (Ten Thousand Persons) | | | Total Water Resources (Billion m³) | | |
|---|---|---|---|---|---|---|---|---|---|
| Year | Real Value | Simulated Value | Error | Real Value | Simulated Value | Errors | Real Value | Simulated Value | Error |
| 2010 | 2976.08 | 2961.07 | −0.0050 | 745.7 | 745.7 | 0.0000 | 30.12 | 31.73 | 0.0534 |
| 2011 | 3642.3 | 3479.96 | −0.0446 | 752 | 796.035 | 0.0586 | 28.33 | 29.42 | 0.0384 |
| 2012 | 4167.98 | 4015.88 | −0.0365 | 757 | 834.483 | 0.1023 | 30.99 | 31.03 | 0.0012 |
| 2013 | 4696.01 | 4562.44 | −0.0284 | 761 | 841.22 | 0.1054 | 29.44 | 30.278 | 0.0284 |
| 2014 | 5250.09 | 5119.51 | −0.0249 | 770 | 853.70 | 0.1087 | 49.69 | 49.69 | 0.0000 |
| 2015 | 5830.95 | 5631.46 | −0.0445 | 779 | 855.91 | 0.0987 | 49.76 | 49.83 | 0.0014 |
| 2016 | 6544.26 | 6222.77 | −0.0491 | 852 | 869.83 | 0.0209 | 87.83 | 90.068 | 0.0254 |
| 2017 | 7366.64 | 6832.6 | −0.0727 | 873 | 871.18 | −0.0021 | 37.64 | 37.64 | 0.0000 |
| 2018 | 8605.13 | 7413.37 | −0.1385 | 893 | 899.31 | 0.0071 | 54.31 | 53.22 | −0.0201 |
| 2019 | 9370.21 | 8046.47 | −0.1413 | 916 | 931.71 | 0.0172 | 21.52 | 20.19 | −0.0618 |
| 2020 | 10,005.56 | 8679.73 | −0.1325 | 937 | 936.69 | −0.0003 | 89.15 | 85.47 | −0.0412 |
| 2021 | 11,412.8 | 9889.62 | −0.1335 | 946.5 | 1177.13 | 0.2437 | 51.06 | 50.68 | −0.0074 |

## 4. Model-Simulation Analysis

After establishing the system-dynamics model of urban green development in Hefei City, as well as completing the construction work of the equations between the variables, this study simulated and analyzed the system-dynamics model. As shown in Figure 5, the simulation results reveal the multidimensional dynamic process of urban development.

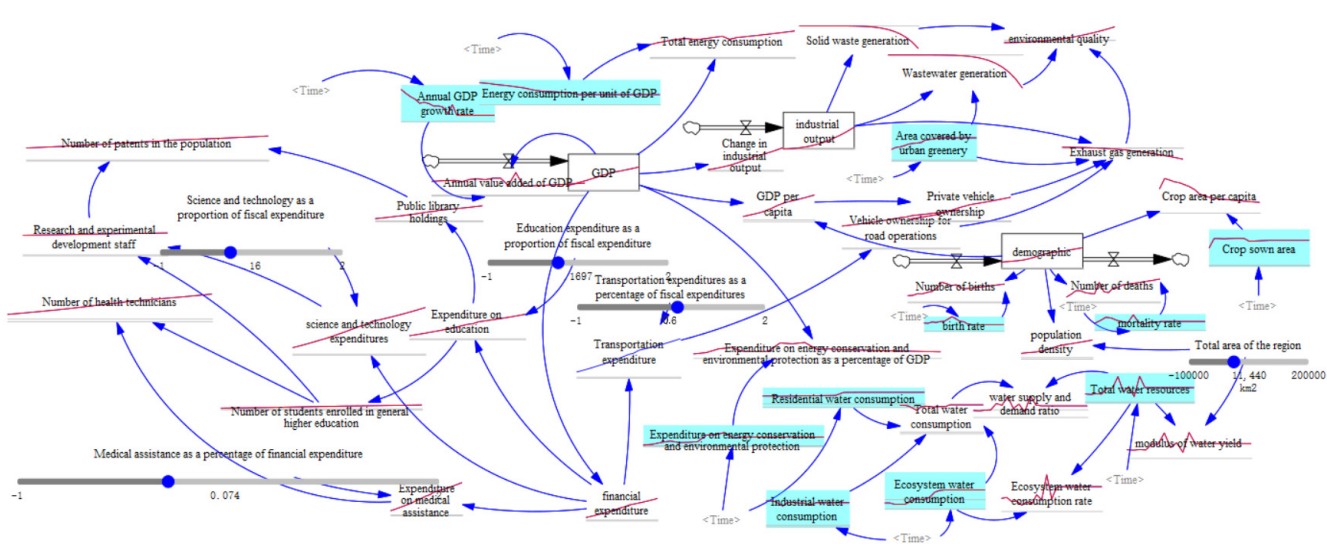

**Figure 5.** Simulation of system dynamics model for green development in Hefei City.

### 4.1. Economic Subsystem Analysis

The simulation results are shown in Figure 6. The GDP of Hefei maintains an upward trend, although the annual growth rate decreases, which fits the general environment of China's gradual slowdown in economic development. This slowdown is caused by a

shift in the pattern of economic development from the high-speed growth of the past to high-quality development. The steady growth in GDP provides the government with more financial resources, which are used to support the construction of various public services, including healthcare, transportation and education.

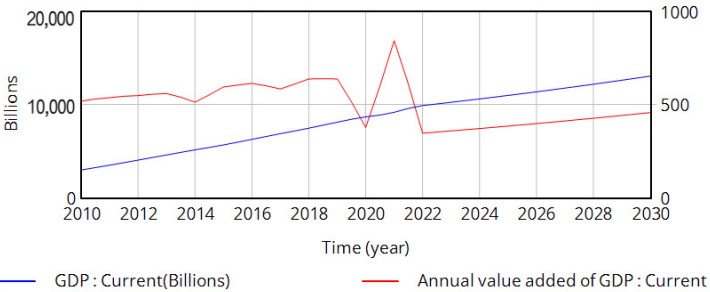

**Figure 6.** Analysis of the simulation results for the economic system.

*4.2. Livelihood Subsystem Analysis*

As shown in Figure 7, the model simulation shows that the livelihood-related indicators are all improved, and Figure 7a–c are the indicators of the livelihood subsystem. Figure 7a represents the level of medical care in people's livelihood, Figure 7b is the level of education and Figure 7c is the level of personal economy. Specifically, the increase in healthcare expenditure has contributed to the growth in the number of health technicians, and the increase in education expenditure has led to a continuous rise in the number of students in higher-education institutions and an increase in the number of books in libraries. In addition, the ownership of private automobiles has increased, reflecting the rising standard of living of the population. These changes indicate that the well-being of the population is increasing with economic growth.

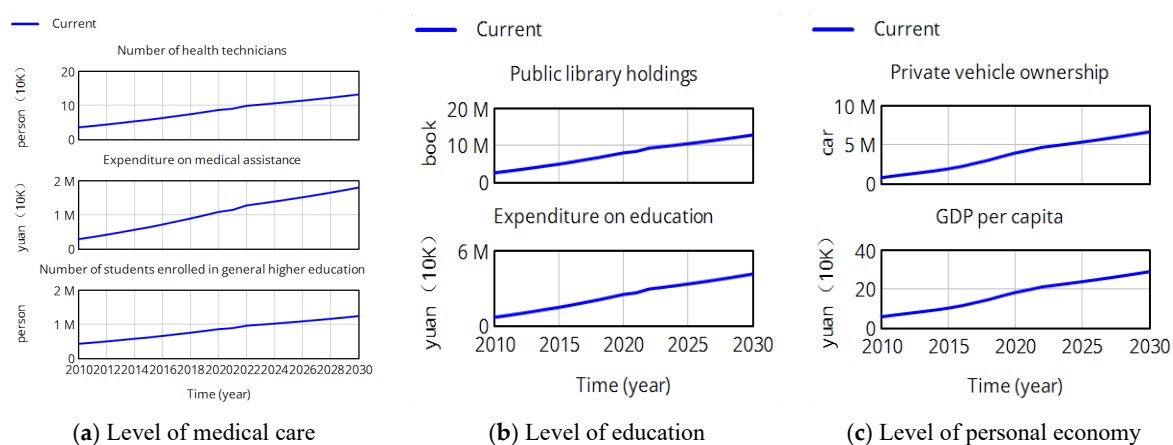

**Figure 7.** Analysis of the simulation results for the livelihood subsystem.

*4.3. Science and Technology Subsystem Analysis*

As shown in Figure 8, the simulation results demonstrate the positive association between S and T investment and urban development. As S and T spending increases, there is a significant increase in the number of students and researchers in higher education institutions. This growth is closely linked to the number of patents granted per 10 K population, indicating an increase in S and T innovation capacity.

Specifically, the growth in S and T spending has led to an increase in research activity, which not only attracts more students into research, but also increases the output of research results. This trend is shown in the simulation model as an increase in the number of academic papers and a rise in the number of patents granted, reflecting the increase in the vitality of science and innovation in Hefei. In addition, the increase in

these scientific research achievements also provides strong support for the city's industrial upgrading and technological innovation, thus promoting the overall process of the city's green development.

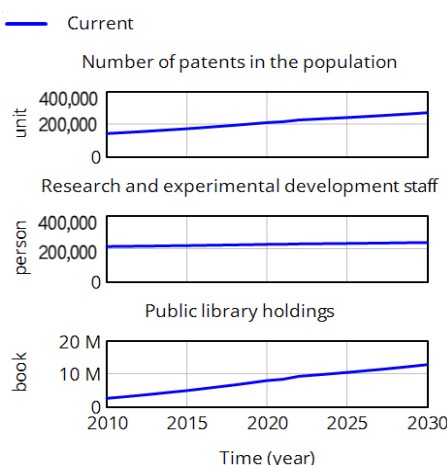

**Figure 8.** Analysis of the simulation results for the technology subsystem.

### 4.4. Environmental Subsystem Analysis

As shown in Figure 9, the simulation results of the environmental subsystem indicate that the total amount of solid waste, wastewater and exhaust gas discharged is on a downward trend, even though urban development is still continuing, as shown in Figure 9a. From the results of the analysis of the livelihood subsystem, it can be seen that private vehicle ownership has been increasing, but the quality of the environment has been improved as well as the emissions having been reduced; this may reflect that the effects of environmental-protection policies are beginning to appear, or the optimization and upgrading of industrial structure, as well as the advancement of environmental management technology. The overall improvement in environmental quality indicates that Hefei is trying to improve its green-development level while realizing economic development. The results also show that the total energy consumption in Hefei City may display an increasing trend in the future, as shown in Figure 9b, but the growth rate may be affected by factors such as energy-structure adjustment, technological innovation and government policies, and may be relatively slow. In general, energy consumption will reduce the environmental quality, but it is obvious that the environmental quality of Hefei will be effectively improved in the future, which indicates that with the continuous progress of science and technology, the efficiency of energy utilization may be improved, which will lead to a gradual decline in energy consumption per unit of GDP in Hefei, as shown in Figure 9c.

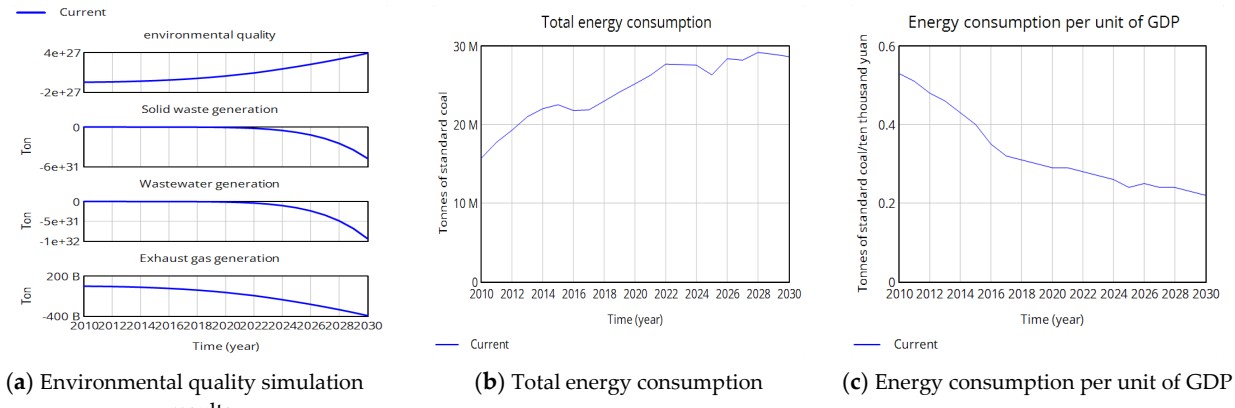

(**a**) Environmental quality simulation results

(**b**) Total energy consumption

(**c**) Energy consumption per unit of GDP

**Figure 9.** Analysis of the simulation results for the environmental subsystem.

In summary, the simulation results show that Hefei City focuses on improving people's livelihood and environmental protection while pursuing economic growth, presenting a city image that is transforming to high-quality and sustainable development, and that the growth in total energy consumption does not represent a decrease in environmental quality. These results provide valuable data-support and decision-making references for future policy formulation, and also provide lessons that can be drawn from for the green development of other cities.

### 4.5. Resource Subsystem Analysis

As shown in Figure 10, the simulation results show that the water consumption in Hefei City will increase year by year in the next few years, and the growth rate may be affected by factors such as water-resource management and the promotion of water-saving technologies. Water resources' availability also has a great impact on water consumption. If water resources are in sufficient supply, it may support the growth of water consumption; conversely, if water resources are in short supply, it may limit the growth of water consumption. The government may take measures to regulate water-use behavior and improve water-use efficiency to ensure the sustainable use of water resources.

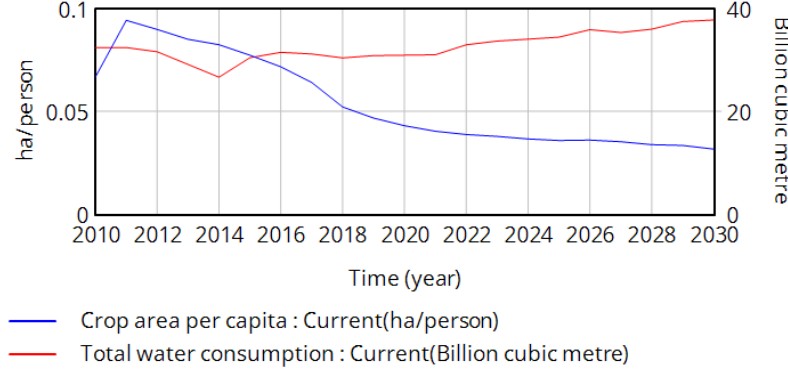

**Figure 10.** Analysis of the simulation results for the resource subsystem.

Regarding arable land resources, the results show that the per capita crop-sowing area in Hefei City may decrease in the future, which means that the arable land resources may face a certain degree of reduction, but at the same time, with urbanization and agricultural modernization, the government may introduce policies to adjust and optimize land resources, and measures to improve the efficiency of arable-land use and to strengthen their protection and management may occur in order to cope with the challenges of land resources and to ensure agricultural production and food security.

## 5. Parameter Analysis

### 5.1. Impact of Changes in Fiscal Expenditure on Livelihood Subsystem

In order to deeply understand the impact of changes in fiscal expenditure on the livelihood subsystem, fiscal expenditure is increased and decreased by a certain percentage, respectively, and its specific impact on the medical and education fields is observed. In the simulation scenario, the proportion of change in fiscal expenditure is referenced to previous changes in fiscal expenditure. After calculation, as shown in Figure 11, except for 2011 and 2012, when the proportion of fiscal expenditure was higher, the increase in fiscal expenditure in the remaining years was basically around 10%, and the reason for the lower increase in fiscal expenditure in recent years may be due to the epidemic.

Therefore, the financial expenditure will be increased and decreased by 10%, respectively in the simulation calculation. As shown in Figure 12a–c represent some important indicators in the livelihood subsystem. The case of keeping the initial state is represented by the green line (original), the case of a 10% increase in fiscal expenditure is represented

by the red line (10% increase), and the case of a 10% decrease in fiscal expenditure is represented by the blue line (10% decrease).

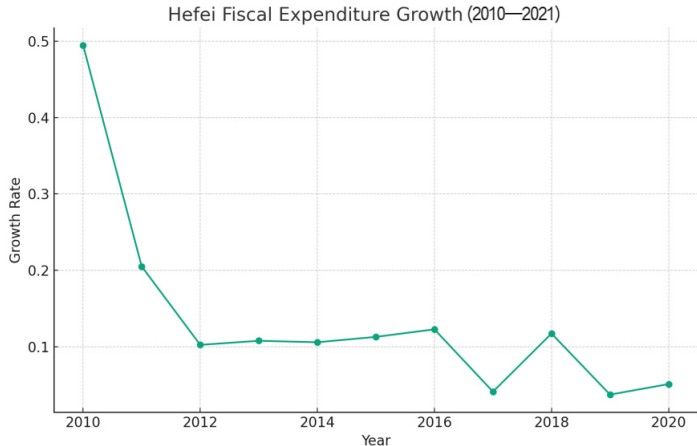

**Figure 11.** Hefei fiscal expenditure growth (2010–2021).

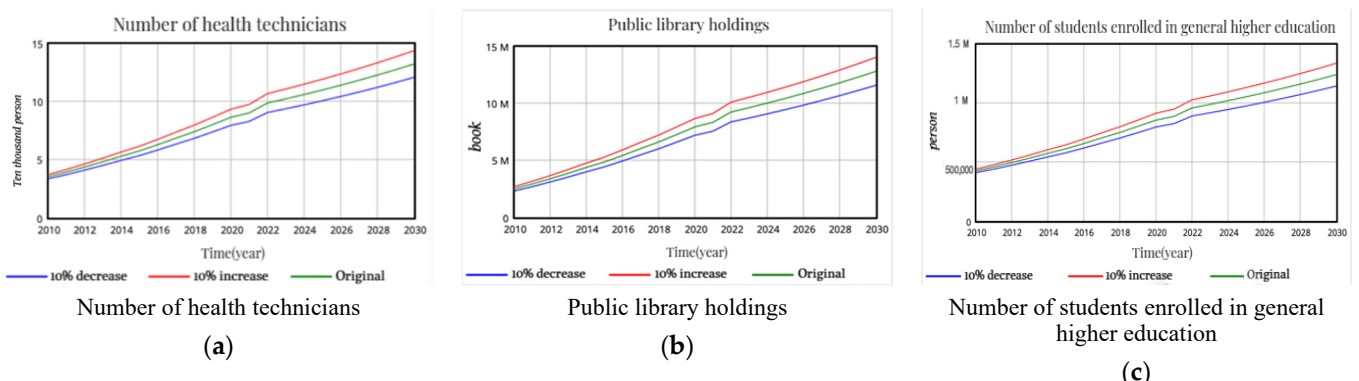

**Figure 12.** Impact of changes in fiscal expenditure on the livelihood subsystems.

In analyzing the results of the simulations, it is noted that an increase in fiscal expenditures results in positive improvements in the healthcare and education sectors. Specifically, in the healthcare sector, an upward trend in the number of health technicians is observed as fiscal expenditure increases, suggesting that more fiscal investment may be used to recruit more health professionals or to improve the level of training and working conditions of existing staff. This may lead to an improvement in the overall quality of healthcare services, and the health status of the population is expected to improve accordingly. A similar trend is shown in the education sector, where increases in fiscal expenditures are accompanied by increases in enrolment in tertiary institutions and in library collections. This reflects the fact that larger fiscal expenditures may have increased investment in educational resources, such as faculties, teaching facilities and academic materials, which, in turn, enhanced the coverage and quality of education. In contrast, when fiscal spending decreased by 10 percent, the relevant indicators in both health and education declined. This suggests that a reduction in fiscal spending may compress the budgets for these essential public services, limiting the scope for their expansion and quality enhancement, which in turn may have a negative impact on the population's health and education level.

Overall, the simulation results clearly show that changes in fiscal expenditures directly affect the key indicators of the livelihood subsystem. Increasing fiscal spending can boost the health and education sectors, while decreasing fiscal spending may have a constraining effect on these sectors. These findings provide valuable data support for policymakers and emphasize the need to consider the long-term impact on livelihoods when formulating fiscal budgets.

### 5.2. Impact of Changes in Percentage of Expenditure on S and T on Technological Development

In this paragraph of the tuning analysis, we explore the impact of the change in the proportion of science and technology expenditure on the science and technology subsystem by increasing and decreasing the proportion of science and technology expenditure by a certain percentage to observe the changes in the relevant indicators. As can be seen in Figure 13, the change in S and T investment increases year by year, so the percentage of change is set at 5% and 10% in the simulation. Different colored lines are used to represent the various situations, such as in Figure 14a,b are the changes in S and T expenditure and the number of research and experimental development (R & E) personnel.

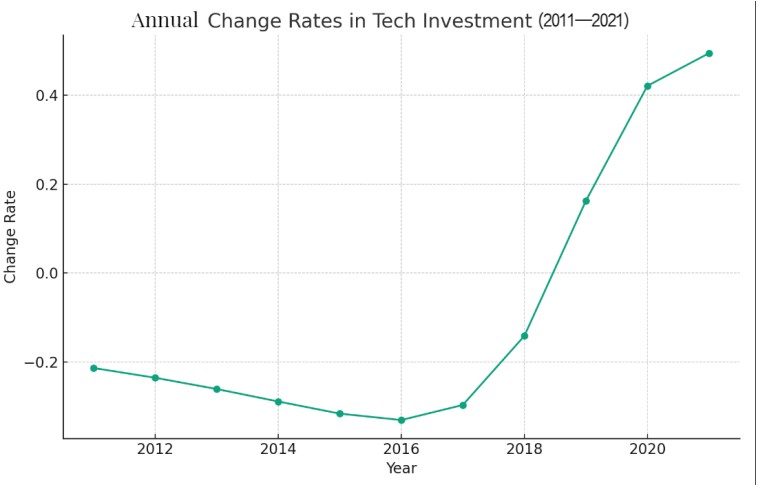

**Figure 13.** Annual change rates in tech investment (2011–2021).

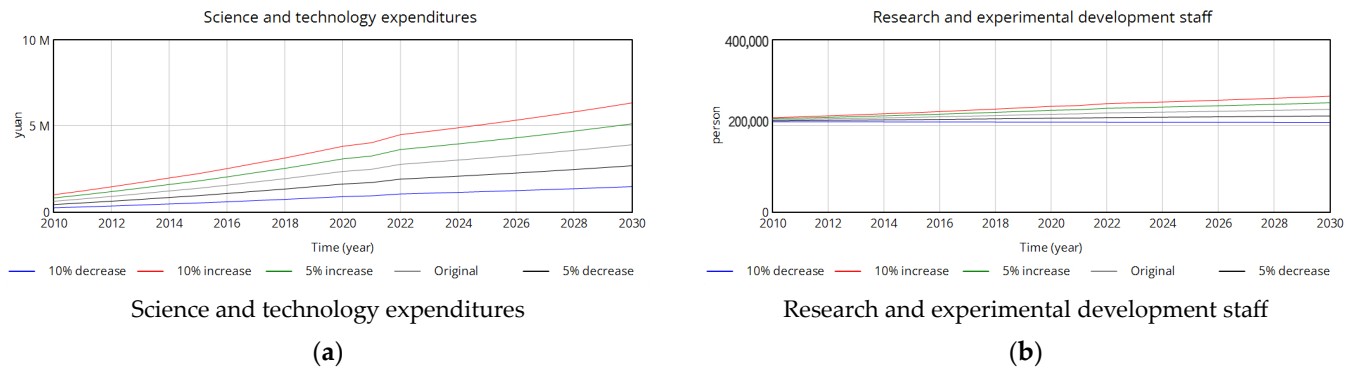

**Figure 14.** Impact of changes in percentage of expenditure on S and T on technological development.

According to the results of the simulations, 5% and 10% increases in the share of S and T expenditures has a positive effect on the size of the research workforce. This is reflected in the increase in the number of researchers and experimental development personnel, which may be due to the fact that the increased funding has allowed for more research projects to be implemented, attracting more students and teachers to participate in research activities. In addition, the increase in the size of the research workforce may have led to an increase in the number of patents granted, which is a direct sign of scientific and technological progress and increased innovation capacity. In the long run, this increase in S and T expenditures may contribute to the accumulation of intellectual property rights, thus enhancing the city's S and T competitiveness and industrial upgrading potential. This is important for promoting the transformation of Hefei into a knowledge-intensive city. Relatively, when S and T expenditures decrease by 5% and 10%, all the above indicators show a downward trend. This suggests that a decrease in S and T expenditure may have a dampening effect on higher education and research activities, affecting the city's long-term

S and T development and innovation capacity. This may not be conducive to the growth of emerging technology enterprises and the transformation of research results.

In summary, the increase or decrease in S and T expenditures has a direct and significant impact on the S and T subsystem of Hefei City. The reasonable adjustment of the proportion of S and T expenditures is not only related to the changes of research output in the short term, but also affects the city's future S and T development strategy and competitiveness layout. The results of these analyses provide data support for the formulation of relevant S and T policies, as well as references for the S and T-development strategies of other cities.

## 6. Conclusions

Urban green development has become an important trend in future urban development and a key way to solve environmental problems, so this study selects Hefei City and applies a system-dynamics approach to construct a comprehensive urban green-development model. We comprehensively examine the performance of five subsystems, namely, economy, livelihood, S and T, environment and resources. In the system-dynamics model of Hefei City, we evaluate the impacts of the changes in fiscal and science and technology expenditures on these subsystems by means of the tuning parameter analysis. The simulation results show that the robust growth of the economic subsystem provides reliable financial support for the development of the other subsystems. In the livelihood subsystem, the increased expenditure on healthcare and education can effectively improve the quality and coverage of public services. The analysis of the science and technology subsystem shows that the increase or decrease in science and technology expenditures directly affects the size of researchers' and innovators' output, thus influencing the city's scientific and technological competitiveness. The simulation of the environmental subsystem shows that environmental-protection measures can effectively alleviate the pressure of urban development on the environment. In the resource subsystem, water consumption may rise year by year, and the per capita sown area of crops may decrease. With the progress of urbanization and the modernization of agriculture, the government may introduce relevant policies to adjust and optimize water and land resources, improve water-use efficiency, and ensure agricultural production and food security. Overall, while pursuing economic development, Hefei City is also focusing on livelihood improvement and environmental protection, showing a city image that is transforming to a high-quality and sustainable development.

In response to some of the system's responses, this study offers the following recommendations: For the economic subsystem, it is recommended that a more flexible fiscal policy be formulated in order to adapt to the new normal of slower economic growth. At the same time, the fiscal-expenditure structure should be optimized to ensure that more funds can be used in areas that promote long-term sustainable development, such as education, healthcare and environmental protection; for the livelihood subsystem, given that increased spending on healthcare and education can significantly improve people's livelihoods, it is recommended that the government continue to increase its investment in these areas and improve the efficiency of public services through policy innovation. In addition, it is recommended to strengthen the regulation of traffic pressure and environmental pollution brought about by the increase in private automobiles. For the science and technology subsystem, considering the important role of science and technology spending on innovation capacity and scientific and technological progress, it is recommended to continue to increase investment in scientific research and promote cooperation between industry, academia and research institutes in order to enhance the rate of transformation of scientific and technological achievements. At the same time, universities and research institutes should be encouraged to cultivate more scientific research talents to support Hefei's S and T-development strategy. For the environmental subsystem, it is recommended that environmental-protection policies be further strengthened, and that the optimization of the industrial structure be promoted to reduce the proportion of highly polluting industries. At the same time, it is recommended that investment in green technology and clean energy

be increased, and that the greening rate of the city be increased in order to improve the quality of the environment and promote the development of a green economy.

Overall, policymakers in Hefei City should fully consider the results of the analysis of the system-dynamics model and comprehensively coordinate the relationships among the subsystems in order to achieve a balanced development of economic benefits, social well-being and environmental sustainability. Through scientific decision making and innovative policy implementation, Hefei can provide a successful example for other cities to transition to high-quality development. The system-dynamics approach has a number of limitations and challenges to forecast, including (1) complexity and uncertainty: many systems are highly complex and uncertain, and their internal interactions and external influences may be difficult to fully understand and model. This complexity and uncertainty make prediction difficult because even if the model performs well at a given point in time, this uncertainty may cause the prediction to fail over time. (2) Data requirements: System-dynamics approaches typically require large amounts of data for parameter estimation and model calibration. However, in some cases, obtaining sufficient data may be difficult or expensive, especially for complex systems or emerging problems. (3) Changes in model parameters: The parameters of many system-dynamics models may change over time; for example, factors such as technological innovations, policy adjustments, or changes in the marketplace may alter the dynamical behavior of the system, thereby affecting the accuracy of the predictions.

**Author Contributions:** Conceptualization, Y.F. and B.L.; methodology, Y.F. and B.L.; Writing—original draft, B.L.; data curation, B.L.; writing—review & editing, Y.F., Q.Y. and G.J.; supervision, Q.Y. and G.J.; funding acquisition, Y.F. and Q.Y. All authors have read and agreed to the published version of the manuscript.

**Funding:** This paper was sponsored by the National Natural Science Foundation of China (No.72101034) and the "new liberal arts program" of the Ministry of Education in China (No.2021090003).

**Data Availability Statement:** The data presented in this study are available on request from the corresponding author.

**Conflicts of Interest:** The authors declare no conflict of interest.

## Appendix A

**Table A1.** Formulas and descriptions of the variables of the Hefei urban green-development system.

| Subsystems | Number | Variant | Formula/Value | Clarification |
|---|---|---|---|---|
| Economic subsystem | 1 | GDP | GDP = INTEG (Annual value added of GDP, 2961.67) | Gross domestic product of Hefei City by year |
| | 2 | Annual value added of GDP | Annual value added of GDP = GDP × Annual GDP growth rate | Annual value added to Hefei's GDP |
| | 3 | GDP growth rate | GDP growth rate = WITH LOOKUP (Time, ([(0, 0)–(2022, 0.2)], (2010, 0.175), (2011, 0.154),(2012, 0.1361), (2013, 0.1221), (2014, 0.1), (2015, 0.105), (2016, 0.098), (2017, 0.085), (2018, 0.0854), (2019, 0.0787), (2020, 0.0434), (2021, 0.092), (2022, 0.035))) | Hefei's annual GDP growth rate |
| | 4 | Industrial output | Industrial output = INTEG (Change in industrial output, 1052.71) | Final results of industrial production activities |
| | 5 | Change in industrial output | Change in industrial output = $100 \times GDP^{1.8}$ | Annual change in industrial output |

**Table A1.** *Cont.*

| Subsystems | Number | Variant | Formula/Value | Clarification |
|---|---|---|---|---|
| Economic subsystem | 6 | Financial expenditure | Financial expenditure = 170 × GDP$^{1.253}$ | Disposal and use of social resources expressed in monetary terms |
| | 7 | Expenditures on science and technology | Expenditures on science and technology = Percentage of science expenditures × financial expenditure | Financial expenditures for science and technology |
| | 8 | Percentage of science expenditures | Percentage of science expenditures = 0.16 | Proportion of fiscal expenditure spent on science and technology |
| | 9 | Transportation expenses | Transportation expenses = Percentage of expenditure on transportation × financial expenditure | Financial expenditures for transportation |
| | 10 | Percentage of expenditure on transportation | Percentage of expenditure on transportation = 0.046 | Percentage of transportation expenditures in financial expenditures |
| | 11 | Expenditure on education | Expenditure on education = financial expenditure × Percentage of expenditure on education | Financial expenditures for education |
| | 12 | Percentage of expenditure on education | Percentage of expenditure on education = 0.1697 | Proportion of fiscal expenditure spent on education |
| | 13 | Expenditure on medical assistance | Expenditure on medical assistance = Percentage of medical assistance × financial expenditure | Financial expenditures for healthcare |
| | 14 | Percentage of medical assistance | Percentage of medical assistance = 0.074 | Proportion of financial expenditures spent on healthcare |
| Livelihood subsystem | 15 | Number of health technicians | $4.779 \times 10^{-6}$ × Expenditure on medical assistance + $2.974 \times 10^{-6}$ × Number of students enrolled in general higher education + 0.905 | All employees of health institutions |
| | 16 | Demographic | Demographic = INTEG (Number of births − Number of deaths, 745.7) | Population of Hefei City in that year |
| | 17 | Birth rate | Birth rate = WITH LOOKUP (Time, ([(0, 0)–(2022, 10)], (2010, 0.1109), (2011, 0.108), (2012, 0.126), (2013, 0.1188), (2014, 0.131), (2015, 0.134), (2016, 0.164), (2017, 0.1994), (2018, 0.1695), (2019, 0.1356), (2020, 0.1287), (2021, 0.0978), (2022, 0.0897))) | Birth rate in Hefei City in that year |
| | 18 | Mortality rate | Mortality rate = WITH LOOKUP (Time, ([(0, 0)–(20, 10)], (2010, 0.0434), (2011, 0.0597), (2012, 0.0797), (2013, 0.0754), (2014, 0.0611) , (2015, 0.053), (2016, 0.0468), (2017, 0.095), (2018, 0.0468), (2019, 0.0384), (2020, 0.0509), (2021, 0.048 ), (2022, 0.0539))) | Population mortality rate in Hefei City in that year |
| | 19 | Number of births | Number of births = demographic × birth rate | Number of births in Hefei in the current year |

**Table A1.** *Cont.*

| Subsystems | Number | Variant | Formula/Value | Clarification |
|---|---|---|---|---|
| Livelihood subsystem | 20 | Number of deaths | Number of deaths = demographic × mortality rate | Number of people who died in Hefei during the year |
| | 21 | GDP per capita | GDP per capita = 0.008 × demographic + 0.001 × GDP − 3.306 | GNP per capita |
| | 22 | Private car ownership | Private car ownership = 253480 × GDP per capita − 701165 | Private car ownership in Hefei City in that year |
| | 23 | Ownership of road-operating vehicles | Ownership of road-operating vehicles = WITH LOOKUP (Transportation expenses, ([(0, 0)–(10, 10)], (12.01, 84,315), (14.485, 103,954), (14.9059, 987,44), (15.1848, 92,741), (17.7759, 97,975), (21.1361, 66,785), (21.6821, 98,099), (24.3429, 98,646), (33.47, 98,953), (36.4077, 92,262))) | Ownership of road-operating vehicles in Hefei in the current year |
| | 24 | Public-library holdings | Public-library holdings = 2.958 × Expenditure on education + 567293 | Public-library collections in Hefei City in that year |
| | 25 | Number of students enrolled in general higher education | Number of students enrolled in general higher education = 0.234 × Expenditure on education + 272490 | Number of students enrolled in ordinary higher-education institutions in Hefei City in the current year |
| Science and technology subsystem | 26 | Number of patents granted to population | Number of patents granted to population = −0.0209624 × Research and experimental development staff − 0.0125769 × Public-library holdings + 115202 | Number of patents granted per 10,000 people in Hefei City |
| | 27 | Research and experimental development staff | Research and experimental development staff = −0.0209624 × Number of students enrolled in general higher education − 0.0125769 × Expenditures on science and technology + 115202 | Number of people engaged in research and experimental development in the city of Hefei |
| Environmental subsystem | 28 | Solid-waste generation | Solid-waste generation = −1.30771 × $2.718 \times$ industrial output$^3$ + 1.06979 × 2.718 − 3 × industrial output$^2$ + 0.972946 × 2.718 + 3 | Solid-waste Generation in Hefei for the Year |
| | 29 | Area covered by urban greenery | Area covered by urban greenery = WITH LOOKUP (Time, ([(0, 0)-(2021, 30,000)], (2010, 12,737), (2011, 14,804), (2012, 15,334), (2013,16,683), (2014, 18,428), (2015, 19,072), (2016, 19,477), (2017, 20,115), (2018, 22,893), (2019, 23,382), (2020, 23,851), (2021, 25,195))) | Covered area of urban greening in Hefei |
| | 30 | Exhaust-gas generation | Exhaust-gas generation = 0.45 × Ownership of road-operating vehicles + 30.565 × Area covered by urban greenery − 19.999 × industrial output − 0.193 × Private vehicle ownership − 291695 | Volume of emissions from Hefei City in the year |

**Table A1.** *Cont.*

| Subsystems | Number | Variant | Formula/Value | Clarification |
|---|---|---|---|---|
| Environmental subsystem | 31 | Wastewater generation | Wastewater generation = $2.718 \times 9.64643 - 7 \times$ industrial output$^3$ $- 7.71311 \times 2.718 - 0.3 \times$ industrial output$^2$ + $1.93174 \times 2.718$ + industrial output $- 9.52156 \times 2.718 + 3$ $-$ Area covered by urban greenery | Amount of wastewater discharged by Hefei City in the year |
| Resource subsystem | 32 | Total water consumption | Total water consumption = Industrial water consumption + Residential water consumption + Ecosystem water consumption | Combined volume of water used by the city of Hefei in one year |
| | 33 | Total energy consumption | Total energy consumption = GDP $\times$ 10000 $\times$ Energy consumption per unit of GDP | Total energy consumption in Hefei in one year |
| | 34 | Crop area per capita | Crop area per capita = Crop-sown area/(demographic $\times$ 10000) | Area of food crops and cash crops actually sown by each individual throughout the year |
| | 35 | Total water resources | Total water resources = WITH LOOKUP (Time, ([(0, 0)–(2021, 10)], (2010, 30.12), (2011, 28.33),(2012, 30.99), (2013, 29.44), (2014, 49.69), (2015, 49.76), (2016, 87.83), (2017, 37.64), (2018, 54.31), (2019, 21.52), (2020, 89.15), (2021, 51.06)) | The sum of surface runoff and infiltration recharge from precipitation. |
| | 36 | Crop-sown area | Crop-sown area = WITH LOOKUP (Time, ([(0, 0)–(2022, 10)], (2010, 497,007), (2011, 751,154), (2012, 750,314), (2013, 743,269),(2014, 751,371), (2015, 754,301), (2016, 755,622), (2017, 754,456), (2018, 678,368), (2019, 68,2262), (2020, 689,089), (2021, 696,104), (2022, 701,900)) | Area of food and cash crops actually sown throughout the year |
| | 37 | Modulus of water yield | Modulus of water yield = Total water resources/Total area of the region | Water resources per unit area of the region |

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
