# Peer review of "Forecasting Research on Urban Green Development Based on System Dynamics—A Case Study of Hefei in China"

_systems, doi:10.3390/systems12040109_

Round 1

Reviewer 1 Report

Comments and Suggestions for Authors

Overall, the article is well structured and interesting. Some minor topics are:

1. Caption figures need to be expanded and be more precise.

2. References need to be updated. There are a few recent ones.

3. Some figures are of low quality

The following points need to be improved significantly:

1. Some predictions seem exaggerated. Others need to make more sense, like the topic of books... nowadays books are replaced by electronic ones. This question needs to be clarified. The fits to the data may not be very precise, and the authors must be cautious with the conclusions. Would you rather talk about trends instead of forecasts? Figure 8 is definitely 'overly optimistic' and could be more understandable.

5. The data, which should be on the website (https://www.hefei.gov.cn/mlhf/sj/nd/), is not available 

Comments on the Quality of English Language

Reviewer 2 Report

Comments and Suggestions for Authors

The research is on urban green development, focusing on a system dynamics model for Hefei City, China. It addresses how urban green development, integrating economic growth, technological innovation, and environmental sustainability, impacts the city's future. The study analyzes the dynamics of four subsystems: economy, livelihood, science and technology, and environment, demonstrating the interplay between economic development and green policies. It provides simulation results, explores the effects of fiscal and scientific investment changes, and offers recommendations for policy adjustments to balance economic, social, and environmental goals. The key findings are:

·       The simulation results reveal the multidimensional dynamic processes of urban development in Hefei City.

·       Economic growth maintains an upward trend despite a slowdown, which aligns with China's shift towards high-quality development.

·       The livelihood subsystem improves with increased healthcare and education expenditures, reflecting enhanced living standards and well-being.

·       The science and technology (S&T) subsystem benefits significantly from increased S&T investment, leading to a rise in research activity, academic output, and innovation.

·       Environmental protection efforts are showing positive outcomes with a decrease in pollutants despite ongoing urban development.

The study shows interconnectedness of the economic, livelihood, S&T, and environmental subsystems. It highlights how fiscal and scientific investments impact these areas and emphasizes the importance of balanced development. Recommendations include adopting flexible fiscal policies, increasing public service investments, regulating private car use, and enhancing environmental protection. The role of S&T spending in fostering innovation and the significance of improving public services like healthcare and education are also discussed.

The paper concludes that Hefei City is on a path towards high-quality and sustainable development, focusing on economic growth, improved livelihoods, and environmental protection. The city serves as a model for balanced urban development. The study suggests that policymakers should consider the simulation results for future planning to ensure economic, social, and environmental sustainability. Recommendations for policy adjustments include optimizing fiscal expenditure, increasing investment in key public services, and strengthening environmental policies.

The paper follows the standard academic structures, including an abstract, keywords, introduction, methodology, results, and conclusions. It also discusses the implications of the findings and provides recommendations for policymakers.

However, the article must better highlight the paper's contribution to existing knowledge, provide a more detailed description of the methodology and analysis, including important international research and literature, add important aspects of green growth such as resource consumption in the model structure and provide a more detailed discussion on the validity of the conclusions. In detail:

·       Specify the research question and highlight novelty of the research.

·       Add a intensive literature review about existing SD models on city development.

·       Integrate the important aspect of resource use such as material, energy, land and water in the model to get a complete picture of the environmental impacts.

·       Add in the causality diagram the information about reinforcing and balancing loops.

·       Provide a more detailed validation and verification of the simulation results.

·       The discussion must better elaborate on the simulation results, e.g. that environmental quality is improved although number of polluters, for example cars, goes up.

Comments on the Quality of English Language

Moderate editing of English language required

Reviewer 3 Report

Comments and Suggestions for Authors

The research theme of forecasting urban green development in Hefei, China, is highly relevant given the current global emphasis on sustainable development. The introduction effectively contextualizes the significance of green development within China's economic strategies and global environmental concerns. The comprehensive review of previous studies establishes a strong foundation for understanding the evolution of green development concepts, highlighting the importance and timeliness of the research theme.

Overall, the text demonstrates a good level of coherence and flow. However, there are instances where the transitions between sections could be smoother. For example, the transition from the introduction to the system dynamics methodology section could be improved to provide clearer linkage between the theoretical background and the chosen methodology. Additionally, the narrative style occasionally becomes dense due to the abundance of technical terminology and complex concepts, which might pose challenges for readers unfamiliar with the subject matter.

While the methodology is appropriate for the research objectives, more emphasis on the limitations and challenges of the chosen approach could enhance the rigor of the study.

There are a few instances where additional recent references could strengthen the discussion, particularly in areas where advancements or alternative viewpoints exist. Ensuring a balance between recent research and foundational literature would enhance the credibility and currency of the study.

Concerning the structue, enhancing the clarity of headings and subheadings could further improve the readability and organization of the document.

The text exhibits generally good grammar and sentence structure. However, there are instances of awkward phrasing and grammatical errors that disrupt the readability. Attention to sentence clarity and coherence, as well as careful proofreading for grammatical accuracy, would enhance the overall quality of the writing.

Round 2

Reviewer 2 Report

Comments and Suggestions for Authors

Can be published in present form. 

Comments on the Quality of English Language

Editing of English language is recommended.
